# The Peanut Skin Procyanidins Attenuate DSS-Induced Ulcerative Colitis in C57BL/6 Mice

**DOI:** 10.3390/antiox11112098

**Published:** 2022-10-25

**Authors:** Na Wang, Weixuan Chen, Chenxu Cui, Yuru Zheng, Qiuying Yu, Hongtao Ren, Zhigang Liu, Chao Xu, Gaiping Zhang

**Affiliations:** 1College of Veterinary Medicine, Henan Agricultural University, Zhengzhou 450002, China; 2College of Food Science and Technology, Henan Agricultural University, Zhengzhou 450002, China; 3International Joint Research Center for Animal Immunology, Zhengzhou 450002, China; 4Key Laboratory of Nutrition and Healthy Food of Zhengzhou, Zhengzhou 450002, China; 5College of Food Science and Engineering, Northwest A&F University, Yangling 712100, China

**Keywords:** peanut skin, procyanidins, DSS-induced, ulcerative colitis

## Abstract

Polyphenols from peanut skin have been reported to possess many beneficial functions for human health, including anti-oxidative, antibacterial, anticancer, and other activities. To date, however, its anti-inflammatory effect and the underlying mechanism remain unclear. In this study, the anti-inflammatory effect of peanut skin procyanidins extract (PSPE) and peanut skin procyanidins (PSPc) were investigated by a dextran sodium sulfate (DSS)-induced colitis mouse model. The results showed that both PSPE and PSPc supplementation reversed the DSS-induced body weight loss and reduced disease activity index (DAI) values, accompanied by enhanced goblet cell numbers and tight junction protein claudin-1 expression in the colon. PSPE and PSPc treatment also suppressed the inflammatory responses and oxidative stress in the colon by down-regulating IL-1β, TNF-α, and MDA expressions. Meanwhile, PSPE and PSPc significantly altered the gut microbiota composition by increasing the relative abundance of *Clostridium XlVb* and *Anaerotruncus*, and inhibiting the relative abundance of *Alistipes* at the genus level. PSPE and PSPc also significantly elevated the production of short-chain fatty acids (SCFAs) in mice with colitis. The correlation analysis suggested that the protective effects of PSPE and PSPc on colitis might be related to the alteration of gut microbiota composition and the formation of SCFAs. In conclusion, the current research indicates that supplementation of PSPE and PSPc could be a promising nutritional strategy for colitis prevention and treatment.

## 1. Introduction

Inflammatory bowel disease (IBD), including Crohn’s disease and ulcerative colitis (UC), begins in the rectum and extends to the proximal colon [1]. The main symptoms are intestinal mucosal injury, relapsing, abdominal pain, diarrhea, mucinous bloody stools, and weight loss [2]. UC is more prevalent than Crohn’s disease globally. Based on current knowledge, it is possible that a combination of genetic and environmental variables contributes to the development of UC, and these modifications lead to abnormalities in the gut microbiome and dysregulation of the mucosal immune system [3]. Pharmacological therapy is used to treat the majority of UC patients. Oral and rectal 5-aminosalycilates are frequently used for mild to moderate UC. Thiopurines, biological agents that target tumor necrosis factor and integrins, as well as small-molecule Janus kinase inhibitors, are among the drug classes used to treat mild to severe colitis. However, up to 15% of individuals will require surgery if medical therapy is ineffective or if dysplasia develops as a result of their protracted colitis [4]. Therefore, it is of great significance to seek new and effective therapies for UC.

Nutritional intervention is an alternative treatment for UC [5]. Numerous studies suggest that polyphenols and their metabolites with potent anti-inflammatory, antioxidant, anticancer, and immunomodulatory properties may be promising contenders for combating UC [6]. In rodent models of colitis, polyphenol-rich diets and substances have been utilized as a therapy, including green tea polyphenols [7], daidzein-rich isoflavones extract [8], polyphenol-enriched cocoa extract [9], curcuma longa extract [10], pomegranate extract and urolithin A [11]. Peanut skin has been shown to be a promising source of phenolic compounds, including phenolic acids, flavonoids, procyanidins, tannic acid, resveratrol, etc. [12]. Peanut skins (testae or seed coat) are the seed coat of the legume family, as a by-product of peanut processing, the output of peanut skins may reach 1 million tons worldwide in recent years [13], most of which are used as animal feed for animal husbandry or wasted [14]. Peanut skin procyanidins (PSPc), which amount to 128 g/kg [15], are one of the most pivotal and functional polyphenols in peanut skin. A study reported that the procyanidins obtained from peanut skins were mainly A-type dimers, different from the B-type dimers that largely exist in grape seeds [16]. PSPc might possess higher bioavailability than grape seed procyanidins [17]. Moreover, according to many in vitro and in vivo studies, PSPc are shown to have many potentially beneficial functions for human health, including anti-oxidative activities, antibacterial, anticancer, hyperlipidemia prevention, anti-allergy, hypoglycemic capability, and inhibiting acrylamide production [18,19,20,21,22,23,24]. However, whether PSPc has a therapeutic effect on UC and its mechanism are still unclear.

Genetic variety, gut barrier failure, proinflammatory cytokine over-release, and gut dysbiosis are all identified as risk factors for UC [25]. Studies have shown that polyphenol-enriched cocoa extract reduces colon damage, with significant reductions in both the extent and the severity of the inflammation as well as in crypt damage and leukocyte infiltration in the mucosa [9]. TNF and IL-6, two inflammatory biomarkers, rose in colitic animals and significantly decreased after treatment with green tea polyphenols [7]. Resveratrol could increase the number of lactic acid bacteria and bifidobacteria, and reduce the number of enterobacteria upon DSS treatment [26]. Interestingly, previous studies also have shown that Peanut skin procyanidins can improve gut barrier integrity, restrain the inflammation reaction and regulate gut microbiota. A study found that peanut skin procyanidins improved the gut barrier integrality by restoring gut morphology and enhancing tight junction protein expression including claudin-1 and occludin in the colon in type 2 diabetic mice [27]. Another study found that both acetone-extracted peanut skin extracts and ethanol-extracted peanut skin extract significantly inhibit COX-2 protein expression, exhibiting similar anti-inflammatory effects [28]. In a study of peanut skin extract to ameliorate atherosclerosis, the results showed that peanut skin extract could significantly change the composition of intestinal microbiota, and its anti-atherosclerosis effect might be related to the changes in the composition and function of intestinal microbiota [29]. Especially, various studies have demonstrated that gut microbiota is crucial in the development of UC [30]. Fecal microbiota transplantation and supplementation with short-chain fatty acids (SCFAs) such as butyrate have been successfully identified as therapies for patients with UC. Although a significant amount of polyphenols are not absorbed along the digestive tract, they may build up in the large intestine, where the intestinal microbiota extensively metabolizes the majority of them. A balanced gut microbiome may play an important role in maintaining human health by producing beneficial microbial metabolites and SCFAs that improve host nutrient supply [31] or by preventing pathogen colonization and maintaining normal mucosal immunity [32]. According to the above, peanut skin procyanidins may exert their therapeutic effects on UC by affecting proinflammatory cytokines and improving gut microbiota diversity and intestinal barrier function.

In the present study, the major components from peanut skin were extracted and analyzed. A dextran sodium sulfate (DSS)-induced colitis mouse model was then used to evaluate the effects of PSPE (peanut skin procyanidin extract) and PSPc (PSPE purified product) regimens on colitis development. By detecting the mucosal damage, junction proteins, inflammatory responses, SCFAs, and gut microbiota diversity, this study explored whether or not the PSPE and PSPc could alleviate UC development by reconstructing the microbiota compositions.

## 2. Materials and Methods

### 2.1. Extraction and Determination of PSPc

The skins of peanuts were provided by Zhengyang Xindi Peanut Group Co., Ltd. (Zhengyang, China), and were frozen at −20 °C until use. PSPE and PSPc were provided by the Key Laboratory of Nutrition and Healthy Food of Zhengzhou, China.

The extraction of PSPE and PSPc was conducted according to the methods optimized in our laboratory. Peanut skin was ground and the oil was removed by petroleum ether (National Pharmaceutical Group Co, Beijing, China), followed by drying, the addition of 70% ethanol (National Pharmaceutical Group Co, Beijing, China), and then extracted jointly by microwave and ultrasonic. The crude extract was separated by Centrifugation (8000 rpm, 5 min) prior to adsorption by activated carbon to attain the filtrate. The concentrated liquor was frozen and dried after extraction, and the resulting PSPE purity was at about 67%.

On the other hand, PSPc were obtained through the following steps. PSPE was chromatographed on macroreticular resins (AB-8, Solarbio Co., Ltd., Beijing, China). The effective ingredients were washed with 40% alcohol, and the elutriant was filtered through a 0.22 µm filter membrane. The PSPc were obtained after the concentrated liquor was frozen and dried. PSPc were determined by HPLC (Thermo Scientific UltiMate 3000 U, Phenomenex Kinetex C18) with a purity of ca. 95%. The composition of PSPc was detected by ultrahigh-performance liquid chromatography–tandem mass spectrometry (UPLC-QTOF-MS/MS) (Agilent Technologies Technology Co., Ltd., Beijing, China). The operating conditions of UPLC-QTOF-MS/MS were as follows: waters BEH C18 column (2.1 × 100 mm, 1.7 µm), mobile phase A: 0.1% formic acid in water; mobile phase B: acetonitrile solution, gradient elution: 0 min, 95% A, 5% B; 30 min, 10% A, 90% B; 45 min, 0% A, 100% B; 50 min, 0% A, 100% B; 51 min, 95% A, 5% B; 60 min, 95% A, 5% B. The real-time flow rate was 0.3 mL/min, the injection volume was 5 µL, the real-time mass spectrometry scanning range was 50~1200 m/z, the sheath gas temp 350 °C, the sheath gas flow 12 L/min, the ESI-mode and the voltage were 3200 V.

### 2.2. Animal Experiment

The male mice (C57BL/6J, 8 weeks old, *n* = 40) were purchased from Xi’an Jiaotong University (Xi’an, Shaanxi, China). Mice in groups of ten were raised in a rectangular cage under a controlled environment (25 ± 2 °C temperature, 50% ± 5% humidity) and were fed with a standard diet (AIN-93) with a 12 h light/dark cycle. The laboratory animal production license number was 81803231.

The mice were divided into four groups randomly: the control group (*n* = 10), the DSS group (*n* = 10), the DSS + PSPE group (*n* = 10), and the DSS + PSPc group (*n* = 10). The control group and the DSS group were treated with physiological saline for 17 days by gavage before euthanasia. The DSS + PSPE group (200 mg/kg of PSPE dissolved in saline) and the DSS + PSPc group (200 mg/kg of PSPc dissolved in saline) were administered with corresponding doses by gavage once daily for 17 days [33,34]. The UC mouse model was established by replacing the drinking water in the DSS group, DSS + PSPE group, and DSS + PSPc group with a 3% DSS solution on day 11, and the mice were treated for 7 days. Mice were sacrificed while being anesthetized with 3.5% chloral hydrate (10 mL/kg). Following sacrifice, the colon, blood, cecal contents, and feces were collected and instantly frozen with liquid nitrogen. The remaining colon tissue was kept at −80 °C for additional biochemical and immunoblot analysis, while a portion of the colon tissues was preserved in 4% paraformaldehyde/PBS (*v*/*v*) for histopathological study at room temperature. Experimental protocols complied with Guidelines for the Care and Use of Experimental Animals: 8th edition (ISBN-10: 0-309-15396-4). The Animal Ethics Committee of Northwest A&F University and the BGI Institutional Review Board on Bioethics and Biosafety (BGI-IRB) authorized the animal experiment procedure.

### 2.3. Disease Activity Index (DAI) Evaluation

The DAI score was used to evaluate the development of UC [35]. The score was recorded starting from the first day of DSS treatment. DAI was evaluated by loss of body weight, blood in stools, and formation of stools. The maximum DAI score was 10, and it was made up of body weight loss (0–4), formation of stools (0–2), and blood in stools (0–4). The mice’s body weight loss was scored as follows: body weight loss of less than 1% received a score of 0, body weight loss of 1–5%, 5–10%, 10–15%, and greater than 15% received scores 1, 2, 3, and 4, respectively. The formation of stools was scored as follows: solid and granular stool received score 0, soft and granular stool received score 1, and loose and showing signs of liquid received score 2. An occult blood test kit (BaSO, Guangdong, China) and the manufacturer’s instructions were used to evaluate the presence of blood in the stool. The results were scored as follows: 0 for no evidence of occult blood; 1 for occult blood that was only weakly expressed; 2 for occult blood that was strongly expressed; 3 for occult blood that was strongly expressed and 4 for occult blood that was strongly expressed and visible.

### 2.4. Colon Histopathology Analysis

The tissues were fixed in 4% paraformaldehyde/PBS (*v*/*v*), embedded in paraffin, and stained with hematoxylin and eosin (H&E). Light microscopy (Olympus, Tokyo, Japan) (×160) was used to examine the histopathologic characteristics of a section of colon tissues with a thickness of 5 µm. The histological scores were assessed by experienced staff in the laboratory according to the extent of infiltration of inflammatory cells and mucosa damage, and these staffers had a lot of hands-on experience, but they were blinded to the samples’ treatments. The scores for the infiltration of inflammatory cells were classified using 5 grades: score 0, none; score 1, infiltrate around crypt bases; score 2, infiltrate in muscularis mucosa; score 3, extensive infiltrate in muscularis mucosa with edema; score 4, infiltration of the submucosa. The scores for the extent of mucosa damage were classified into 4 grades: score 0, intact mucosa; score 1, ≤1/3 disruption of mucosa; score 2, 1/3–2/3 disruption of mucosa; score 3, >2/3 disruption of the mucosa. The sum of the two scores was considered the histological score.

### 2.5. Alcian Blue Staining

Goblet cells and mucus layers have frequently been visualized using alcian blue staining. [36]. The tissue was embedded vertically in paraffin and cut into 5 μm sections. Sections were stained with Alcian blue/Nuclear Fast Red. The colon tissue sections were observed at ×160 magnification.

### 2.6. Immunohistochemistry Analysis

The immunohistochemistry (IHC) analysis followed the guidelines previously described [37]. The colon tissue sections were dewaxed, washed three times with PBS, and treated in 3% hydrogen peroxide for 10 min. Then sections were blocked for 20 min by a normal goat serum blocking solution. After blocking, sections were incubated with primary antibody (Anti-Claudin 1 antibody, Abcam, Cambridge, UK, 5µg/mL) overnight at 4 °C. After rinsing with PBS 3 times, sections were incubated with 50 μL secondary antibody (Biotin conjugated Goat Anti-Mouse IgG, ZSGB-Bio, Beijing, China) for 30 min and then was coupled with horseradish peroxidase for 30 min at room temperature. Tissues were rinsed with PBS 3 times before visualizing by chromogen DAB (DAB kit, Zhongshan Golden Bridge Biotechnology Co., Ltd., Beijing, China). The final steps were clearing colon sections in xylene, dehydrating them in ethanol, and mounting them with Permount TM Mounting Medium.

### 2.7. ELISA Assay

Colonic tissues were homogenized and centrifuged, and the supernatant was taken for cytokine analysis. Proinflammatory cytokine concentrations (MDA, TNF-, and IL-1) were quantified by using ELISA kits (Xinle Biotechnology, Shanghai, China), and measured at an optical density of 450 nm (Bio-Rad, Hercules, USA).

### 2.8. qRT-PCR

The protocol of the qRT-PCR was as previously described with the following modifications [38]. The total RNA of the colon tissue was extracted with TRIzol (Jingcai Biotechnology, Xi’an, China) reagent and quantified using NanoDropOne (Thermo Fisher Scientific, MA, USA). The cDNA samples (50 ng/μL) were synthesized using the HiFiScript cDNA Synthesis Kit by RNA reverse transcription. Subsequently, a Real-Time PCR reaction system was prepared by UltraSYBR Mixture (Cowin Biosciences, Beijing, China) for quantitative analysis. The reaction mixture was made by mixing 1 μL upstream primer (6 μM), 1 μL downstream primer (6 μM), 2 μL diluted five-fold sample (10 ng/μL), 6 μL ddH_2_O, and 10 μL mixture (10×). The primers used in qRT-PCR were listed in Table 1, which were laboratory designed and cited in previous research [38]. The level of the RNA expressions was quantified using real-time detection, performed by CFX96™ real-time system (Bio-Rad, Hercules, CA, USA). Thermal cycling conditions were as recommended by the manufacturer for 40 cycles. Cycle threshold (Ct) values were recorded. Data were normalized using GAPDH and transformed using the 2^−ΔΔCT^ method.

### 2.9. Analysis of 16S rRNA Sequencing

The cecal contents samples were collected from mice in a sterile environment after being sacrificed. The 16S rRNA sequence analyses of the collected samples were conducted by the method reported previously [31]. The ribosomal RNA gene’s 16S rDNA V3-V4 region was amplified by PCR using the primers 341F: CCTACGGGNGGCWGCAG; and 806R: GGACTACHVGGGTATCTAAT. Using the AxyPrep DNA Gel Extraction Kit from Axygen Biosciences, amplicons were extracted from 2% agarose gels, purified, and quantified using QuantiFluor-ST (Promega, U.S.). Amplicons were paired-end sequenced (2 × 250) on an Illumina MiSeq platform.

Quantitative Insights Into Microbial Ecology was used to demultiplex, quality-filter, and evaluate the raw 16S rRNA gene sequence data (QIIME). Using UPARSE, the sequences with a similarity of less than 97% were grouped into operational taxonomic units (OTUs). By using the RPD classifier (version 2.2), which is based on the SILVA Database, the representative sequences were divided into organisms. The β diversity and relative abundance of microorganisms were compared using the Kruskal–Wallis ANOVA test. All data were presented in the text as the means ± SE, and *p*  <  0.05 was considered to be a significant difference.

### 2.10. SCFAs Content in Feces

The concentrations of SCFAs in feces were determined with a gas chromatograph (Shimadzu Corporation, GC-2014C, Kyoto, Japan)as reported previously [39]. The standards of SCFAs (Aladdin Bio-Chem Technology Co., Ltd., Shanghai, China) were propionate (P110445), butyrate (B11se0438), isobutyrate (I103524), and isovalerate (I108280). Approximately 200 mg of the fecal content sample was homogenized with 1 mL of distilled water; then, 0.15 mL of 50% H2SO4 (*w*/*w*) and 1.6 mL of diethyl ether were added. After the samples were incubated at 4 °C for 20 min, they were centrifuged at 7155 g for 5 min, and then collected and filtered the organic phase. The conditions were used for GC: a temperature of 50 °C was started, kept for 3 min, then increased to 130 °C at 10 °C/min, 170 °C at 5 °C/min, 220 °C at 15 °C/min, and held at 220 °C for 3 min. The injector and detector had respective temperatures of 250 °C and 270 °C.

### 2.11. Statistical Analysis

Except for 16S rRNA sequencing data, other data were reported as mean  ±  SE, and the significant differences between the means were determined by one-way ANOVA followed by the Newman–Keuls multiple comparison post-test method using GraphPad Prism 6.0 software (GraphPad Software Inc., San Diego, CA, USA). The Kruskal–Wallis H test, a nonparametric test method, was used to evaluate the 16S rRNA sequencing data in order to find the significant differences between the several groups. Means were considered to be statistically significant if *p* <  0.05.

## 3. Results

### 3.1. Composition of PSPc

The composition of extracted and purified peanut skin procyanidins was analyzed by UPLC-QTOF-MS/MS based on the different cleavage modes of procyanidins, combined with information of [M-H]-(m/z) and corresponding fragment ions. The total ion flow diagram of the PSPc is shown in Figure 1, and the secondary mass spectrometry of each compound is shown in Appendix A. The specific analysis of the main substances according to the secondary mass spectrometry is shown in Table 2. PSPc are mainly characterized by A-type procyanidins dimer, Protocatechualdehyde and Catechins; it also includes very little A-type procyanidins trimer, A-type procyanidins tetramer, Protocatechuic acid, and B-type procyanidins.

### 3.2. Effects of PSPE and PSPc Supplementation on Pathological Changes in DSS-Induced Colitis Mice

The mice were treated with PSPE and PSPc for 17 days (Figure 2A). Compared with the control group, the body weight loss was more severe in the DSS models on the end day of DSS treatment (*p* < 0.01) (Figure 2B,C). Although there was no significant difference in the DSS models (Figure 2C), the DAI score was significantly elevated in the DSS-treated mice compared with the control group (Figure 2D). Compared with the DSS group mice, DAI values were reduced in both DSS + PSPE and DSS + PSPc groups (23.0% and 25.3%, respectively) (Figure 2E). Thus, the PSPE and PSPc supplementation exhibited a beneficial effect on alleviating UC development in mice.

### 3.3. Effect of PSPE and PSPc Supplementation on the Inflammatory Factor Expressions and Biomarker of Oxidative Stress in DSS-Induced Colitis Mice

The cytokines IL-1β, TNF-α, and IL-6 play essential roles in intestinal epithelial cells and inflammatory cells, and the overexpression of COX-2 and iNOS may be involved in the occurrence and development of UC [40]. The protein levels of IL-1β and TNF-α in colon tissue were evaluated by ELISA assays. The level of the RNA expressions was quantified using RT-PCR. Compared with the control group, the IL-1β, TNF-α, and IL-6 protein levels were significantly upregulated in the DSS group. However, compared with the DSS group, the protein expressions of IL-1β, TNF-α and IL-6 were significantly down-regulated in the DSS + PSPE and DSS + PSPc groups (Figure 3A–C). Compared with the control group, the mRNA expressions of the inflammatory mediators, including COX-2, and iNOS, were significantly elevated in the DSS group. However, the mRNA expressions of COX-2 and iNOS were significantly lower in the PSPE and PSPc treatment in DSS-treated mice. (Figure 3D,E). The level of MDA in colon tissues, as a biomarker of oxidative stress, has also been detected [41]. The PSPE and PSPc treatment significantly down-regulated the DSS-increased levels of MDA (Figure 3F). These results indicated that PSPE and PSPc supplementation reduced inflammatory factor expressions and suppressed oxidative stress in DSS-induced colitis mice.

### 3.4. Effects of PSPE and PSPc Supplementation on the Histopathological Changes in the Colon of the DSS-Induced Colitis Mice

The H&E staining showed that, compared with the control group, the DSS treatment caused severe colon injury, crypt structure damage, goblet cell loss, massive inflammatory cell infiltration, muscularis thinning, and cortical destruction (Figure 4A). These were typical symptoms of UC development. However, the colon tissue damage induced by DSS was significantly reversed in the DSS + PSPE and DSS + PSPc groups (*p* < 0.05) (Figure 4B). Moreover, Alcian blue staining was employed to detect the number of goblet cells (Figure 4A). The loss of goblet cells was significantly attenuated in the DSS + PSPE and DSS + PSPc groups (Figure 4C). Compared with the DSS group, the DSS-induced mucosal damage was significantly suppressed, and the down-regulated MUC-2 mRNA expression was increased in the DSS + PSPc group (*p* < 0.01) (Figure 4D).

Claudin-1 protein, concentrated in epithelial tissue, plays an essential role in forming impermeable barriers [42]. The IHC analysis shows that reducing Claudin-1 expression in the DSS-treated mice was prevented in the DSS + PSPE and DSS + PSPc groups (Figure 4A,E). Meanwhile, compared with the DSS group, the mRNA expressions of tight junction proteins, including Claudin-1 and occludin were significantly increased in PSPE and PSPc-treated colitis mice (*p* < 0.05) (Figure 4F,G). These results indicated that PSPE and PSPc treatment effectively alleviated the inflammation, epithelial layer damage, goblet cell loss, and Claudin-1 loss in the colon of the colitis mice.

### 3.5. Effects of PSPE and PSPc Supplementation on the Gut Microbiome Diversity in DSS-Induced Colitis Mice

In order to investigate the effects of PSPE and PSPc treatment on the gut microbiome composition, the 16S rDNA sequencing analysis was used in this study. As shown in the principal co-ordinates analysis (PCoA), a significant separation was observed among the Con, DSS, DSS + PSPE, and DSS + PSPc groups (Figure 5A). The linear discriminant analysis (LDA) revealed that *Olsenella* was enriched in the DSS group. Meanwhile, the microbes including *Turicibacter, Pseudomonadales, Moraxellaceae, and Acinetobacter* were enriched in the DSS + PSPE group, and the microbes including *Bacteroides, Bacteroidaceae, Gammaproteobacteria, Cupriavidus, Burkholderiaceae, Clostridium_XVIII* were enriched in the DSS + PSPc group (Figure 5B).

The relative abundances of *Clostridium XlVb* and *Anaerotruncus* increased in the DSS + PSPc group compared with the DSS group (*p* < 0.05). Compared to the control group, the relative abundances of *Alistipes* in the DSS groups increased significantly (*p* < 0.05), whereas the relative abundances decreased in the DSS + PSPE group (Figure 5C–E).

### 3.6. Effect of PSPE and PSPc Supplementation on the SCFAs Generation in DSS-Induced Colitis Mice

Many studies have shown that SCFAs, especially butyrate, have therapeutic effects in patients with IBD [43]. Compared with the DSS group, butyrate, isobutyrate, and isovalerate contents were significantly promoted in the DSS + PSPE and DSS + PSPc groups (Figure 6A–C). The DSS group significantly increased propionate formation in feces compared with the CON group. However, propionate content significantly decreased in the DSS + PSPE and DSS + PSPc groups (Figure 6D). PSPE and PSPc treatment altered SCFAs production in the colon.

### 3.7. Correlation Analysis among the DAI, Fecal SCFAs Levels, Colon IL-1β, TNF-α, MDA Level, and Gut Microbiota

To clarify the correlation between gut microbiota, fecal SCFAs level, and DAI score, Spearman’s correlation analysis was implemented based on these experimental data (Figure 7). The body weight loss was positively correlated with the DAI on the 17 (r = 0.74, *p* < 0.01) but negatively correlated with the fecal isobutyrate (r = −0.58, *p* < 0.05), isovalerate (r = −0.68, *p* < 0.05). The DAI on the 17 was positively correlated with the MDA levels (r = 0.58, *p* < 0.05) but negatively correlated with the fecal propionate levels (r = −0.66, *p* < 0.05). The colonic IL-1β, TNF-α, and MDA levels were decreased in DSS + PSPE and DSS + PSPc groups. The colonic IL-1β levels showed a negative correlation with the fecal butyrate (r = −0.55, *p* < 0.05), but showed a positive correlation with the colonic TNF-α (r = 0.89, *p* < 0.01) and MDA levels (r = 0.76, *p* < 0.01). The colonic TNF-α levels showed a negative correlation with the fecal butyrate (r = −0.72, *p* < 0.01), but showed a positive correlation with the colonic MDA levels (r = 0.67, *p* < 0.05). The fecal propionate levels were positively correlated relative abundance of *Anaerotruncus* (r = 0.63, *p* < 0.05). The fecal butyrate levels were positively correlated relative the fecal isobutyrate (r = 0.96, *p* < 0.01), isovalerate (r = 0.79, *p* < 0.01). The fecal isobutyrate levels were positively correlated the fecal isovalerate (r = 0.79, *p* < 0.01). These results suggested that the beneficial effects of PSPE and PSPc on UC development were related to the balance of gut microbiota compositions and the formation of microbial metabolites.

## 4. Discussion

In the present study, we found that the major components of PSPc are A-type procyanidin dimers, catechins, protocatechualdehyde, etc. Moreover, it has been found that PSPE and PSPc supplementation can reduce DSS-induced body weight loss and DAI value in the colitis mice. PSPE and PSPc supplementation could also reduce intestinal inflammation, epithelial layer damage, goblet cell loss, and elevate the tight junction protein Claudin-1 expression. The PSPE and PSPc treatment significantly down-regulated the levels of inflammatory mediators and the biomarker expression of oxidative stress. Furthermore, the PSPE and PSPc treatments reshaped the gut microbiome compositions and enhanced the generation of microbial metabolite SCFAs in the feces of colitis mice. The correlation analysis demonstrated that the inflammatory responses were highly correlated with the gut microbiota compositions and microbial metabolite formation.

Peanut skin procyanidin extract (PSPE) has been reported to possess various biological activities [22,44,45,46,47,48,49,50,51]. Peanut skin contains a variety of dietary polyphenols, especially procyanidins, which may be major active components of peanut skin [29] and are composed of flavan-3-alcohol structural units, and the monomers, mainly including catechin, epicatechin, catechin gallate, epicatechin gallate, and epicatechin gallate, etc. [52]. The procyanidins can be divided into type A and type B procyanidins according to the different positions of chemical bonds between monomers. Type B procyanidins are formed by C4-C8 or C4-C6 phase linkage of catechin or/with epicatechin monomer and are widely distributed in plants [53]. Type A procyanidins are linked by C2-O-C7 or C2-O-C5 in addition to C4-C8 or C4-C6, which are mainly found in peanuts and cranberries. It has been reported that type A procyanidins which are more stable with higher bioavailability were the main ingredient in peanut red skins [54]. In addition, according to the degree of polymerization between monomers, they can be divided into a dimer, trimer, tetramer, etc. [55]. However, the bioactivity of procyanidins is inversely proportional to the degree of polymerization—the lower the degree of polymerization, the higher the bioactivity of Procyanidins [56]. In this study, PSPc were obtained after purification of PSPE, it was found that PSPc were mainly composed of A-type procyanidins dimer, A-type proanthocyanidin trimer, A-type proanthocyanidin tetramer, catechins, and protocatechualdehyde, etc. These active components suggested that PSPc had better biological activities.

UC is usually accompanied by weight loss, inflammation, oxidative stress, intestinal barrier disruption, and other phenomena. We found that, compared with the control group, DSS-treated mice showed significantly severe body weight loss, significantly elevated the DAI score, and significantly increased the protein expressions of IL-1β and TNF-α, the mRNA expressions of IL-6, COX-2, and iNOS, and the protein expression of MDA in colonic tissue, which showed that the DSS-treated mice had typical symptoms of UC. The intake of polyphenols has been shown to be effective in relieving the symptoms of UC [7,8,9,10,11]. A-type procyanidins could improve the intestinal barrier effect, reduce the expression levels of inflammatory cytokines (TNF-α, IL-β, IL-6, and IL-10) and oxidative stress (MDA, T-SOD, NO, and iNOS) in DSS-induced UC mice [57]. Catechins could significantly inhibit excessive oxidative stress through direct or indirect antioxidant effects, hence promoting activation of the anti-oxidative substances and reducing oxidative damage to the colon [58]. Protocatechualdehyde could play a significant anti-inflammatory role by regulating the activation or inactivation of inflammatory and oxidative stress-related cell signaling pathways, such as NF-κB, MAPKs, and STAT1/3 pathways [59,60]. In the current study, we found that, compared with the DSS group, both PSPE and PSPc supplementation could reduce the DAI values, decrease the levels of TNF-α and IL-1β, decrease the mRNA expressions of IL-6, COX-2, and iNOS, down-regulated the levels of MDA, and increase the levels of tight junction protein, MUC-2 and goblet cells in colitis mice. These results indicated that PSPE and PSPc could improve the symptoms of colitis mice. In addition, PSPc is mainly composed of A-type procyanidins, catechins, protocatechualdehyde, which may play a major role in alleviating ulcerative colitis.

Dysbiosis of the gut microbiome is increasingly considered to be causatively related to UC [30]. Consistent with previous studies, the gut microbiota was dysregulated, with Firmicutes reduced and Bacteroidetes increased in the DSS treatment group. It was accompanied by a loss of diversity in the microbiota [61]. PSPE-treated and PSPc-treated mice had a good protective effect on the intestinal flora of colitis mice. The results showed that *Olsenella*, which observed a significant increase in patients with UC [62], was enriched in the DSS group. The microbes including *Turicibacter, Pseudomonadales, and Moraxellaceae* enriched in the PSPE group have been reported negatively correlated with inflammatory factors and significantly associated with great immune recovery [63,64,65]. Both *Bacteroides* and *Clostridium_XVIII*, enriched in the DSS + PSPc group, were reported to have anti-inflammatory action. Moreover, the abundance of *Clostridium XlVb* and *Anaerotruncus* was significantly lower in the DSS group than in the control group. PSPc treatment significantly increased the level of *Clostridium XlVb* and *Anaerotruncus* in colitis mice.

Previous studies have found that SCFAs are the most important metabolites of gut microbiota and play a very important role in colitis, the fecal SCFAs levels were reduced in active UC [37,38]. In our study, the results showed that the SCFAs, such as butyrate, isobutyrate, and isovalerate, were significantly decreased in the DSS group compared with the control group. The PSPE and PSPc treatment significantly increased the levels of butyrate, isobutyrate, and isovalerate in DSS-treated mice. We conducted a correlation study and found that the fecal butyrate levels were positively correlated with the fecal isobutyrate and isovalerate levels, but negatively correlated with the IL-1β and TNF-α levels. Meanwhile, it has been reported that butyrate is a key metabolite of UC, which regulates immune responses by inhibiting the release of proinflammatory cytokines and enhances intestinal barrier function by upregulating MUC2 expression in UC [43,66].

Previous studies have detected that the genus of *Anaerotruncus*, commonly found in the intestinal tracts of healthy humans and animals, can produce acetic and butyric acids through sugar metabolism pathways [67]. PSPE treatment significantly increased the level of *Anaerotruncus**,* it suggested that the peanut skin procyanidins may attenuate DSS-induced ulcerative colitis by increasing SCFAs content, such as butyrate, isobutyrate, and isovalerate. Despite the multiple beneficial effects of SCFAs on host gastrointestinal activity, excessive quantities of propionic acid have been reported in gingival inflammation, acne, and irritable bowel syndrome [68]. In this study, we found that propionic acid increased in the DSS group compared with the control group, and supplementation of PSPc significantly decreased propionate content in DSS-induced colitis mice. Meanwhile, It has been found that the abundance of *Alistipes,* one of the mainly propionate producers, highly related to dysbiosis and chronic intestinal inflammation [69,70], increased in DSS-induced colitis mice and decreased in the DSS + PSPE group. Based on these results, we suggest that PSPE and PSPc had beneficial effects on alleviating UC development by reconstructing the microbiota compositions and changing the generation of SCFA in a DSS-induced colitis mouse model. In the future, we need to further confirm the effects of PSPE and PSPc on gut microbial metabolism changes and the underlying mechanism involved.

## 5. Conclusions

Taken together, the present results demonstrated that both PSPE and PSPc supplementation significantly ameliorated DSS-induced colitis by alleviating colonic inflammation, inhibiting colonic injury, and maintaining the integrity of the intestinal barrier. In addition, PSPc, mainly A-type procyanidins, could regulate the metabolism by maintaining gut microbiota. It suggests that purified PSPc have potential as dietary supplements or for the production of fortified foods as a promising nutritional strategy for reducing UC.

## Figures and Tables

**Figure 1 antioxidants-11-02098-f001:**
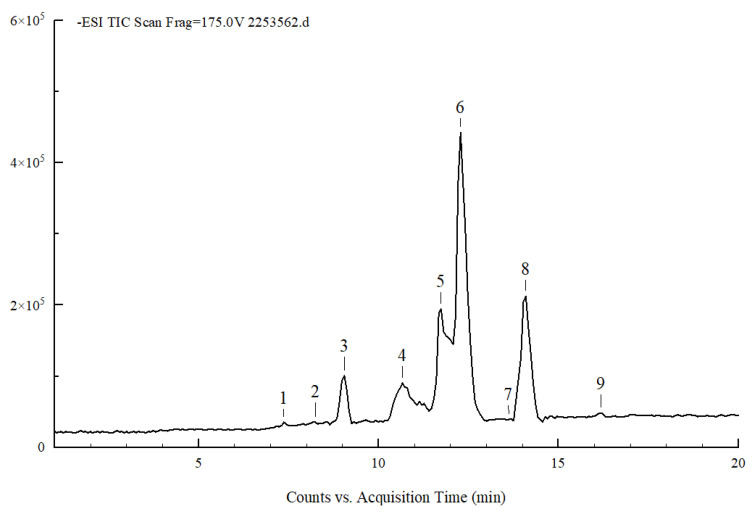
Total negative ion chromatogram of PSPc.

**Figure 2 antioxidants-11-02098-f002:**
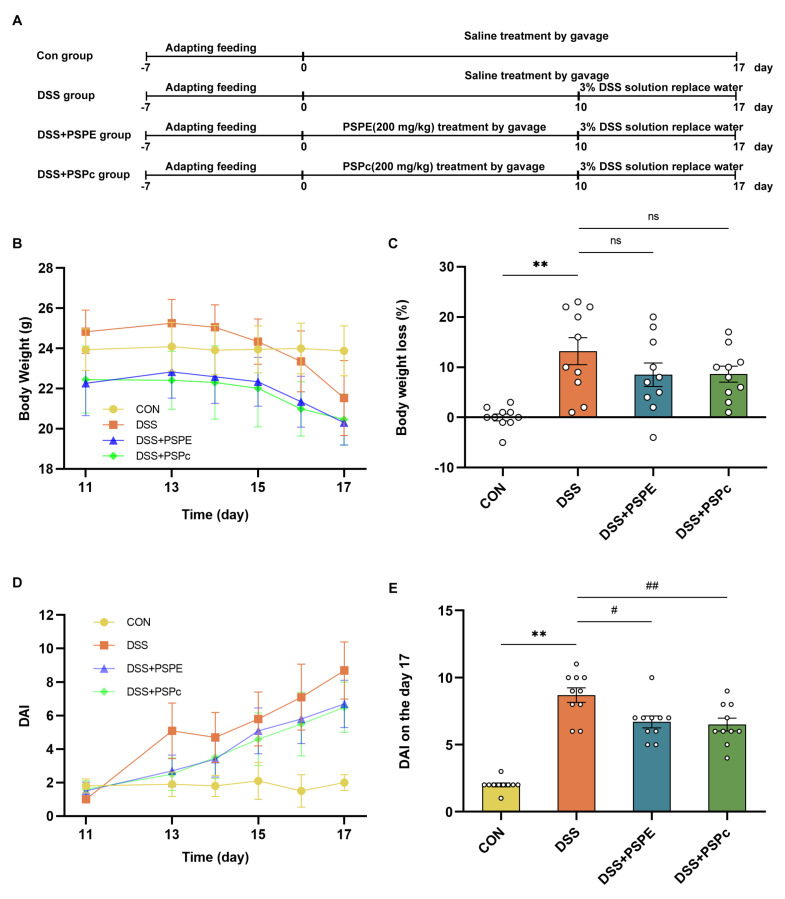
Effect of PSPE and PSPc on the symptoms in the colon tissue of DSS−induced colitis mice. (**A**) Con group: control group (*n* = 10 mice); DSS group: DSS alone group (*n* = 10 mice); DSS + PSPE group: PSPE 200 mg/kg + DSS group (*n* = 10 mice); DSS + PSPc group: PSPc 200 mg/kg + DSS group (*n* = 10 mice). Schematic representation of the experimental design; (**B**) body weight during DSS treatment (11th–17th day); (**C**) body weight loss during the DSS treatment ([body weight on 11th day-body weight on 17th day]/body weight on 11th day); (**D**) DAI score during DSS treatment; (**E**) DAI on the seventh day after DSS-induced UC. Data were expressed as mean ± SE (*n* = 3 mice). ** *p* < 0.01, compared with the Con group, # *p* < 0.05, ## *p* < 0.01, compared with the DSS group, ns = not significant.

**Figure 3 antioxidants-11-02098-f003:**
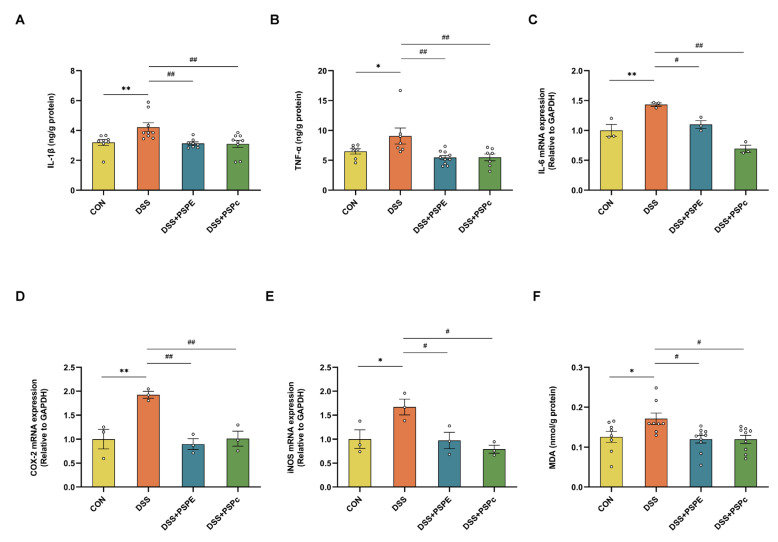
Effect of PSPE and PSPc on the proinflammatory cytokines and mRNA expression in colonic tissue. (**A**,**B**) Proinflammatory cytokines (IL-1β and TNF-α) in colonic tissue. Data were expressed as mean ± SE (*n* = 7–10 mice). (**C**–**E**) qRT-PCR expression (IL-6, COX-2, and iNOS). Data were expressed as mean ± SE (*n* = 3 mice). (**F**) MDA in colonic tissue. Data were expressed as mean ± SE (*n* = 7–10 mice). * *p* < 0.05, ** *p* < 0.01, compared with the Con group, # *p* < 0.05, ## *p* < 0.01, compared with the DSS group, ns = not significant.

**Figure 4 antioxidants-11-02098-f004:**
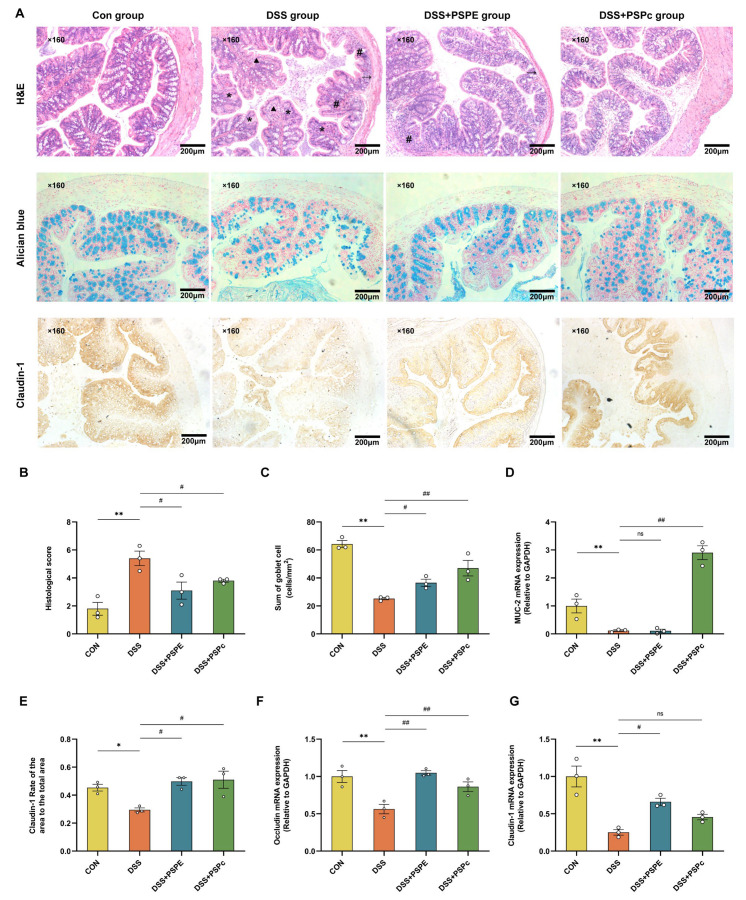
Effect of PSPE and PSPc on the histopathological injury and gut barrier integrity in colon tissue of DSS-induced colitis mice. (**A**) Representative images of H&E staining, Alcian blue staining, and claudin-1 protein IHC staining of the colon for each group. * damaged crypts structure, # loss of goblet cells, ▴ massive infiltrating inflammatory cells, → muscularis thinning. Scale bar: 200 μm (×160); (**B**) histopathology scores of colonic tissues; (**C**) numbers of goblet cells in the colon; (**D**) qRT-PCR expression (MUC-2); (**E**) quantitative analysis of claudin-1 protein content in colonic tissue; (**F**,**G**) qRT-PCR expression (Claudin-1 and Occludin). Data were expressed as mean ± SE (*n* = 3 mice). * *p* < 0.05, ** *p* < 0.01, compared with the Con group, # *p* < 0.05, ## *p* < 0.01, compared with the DSS group, ns = not significant.

**Figure 5 antioxidants-11-02098-f005:**
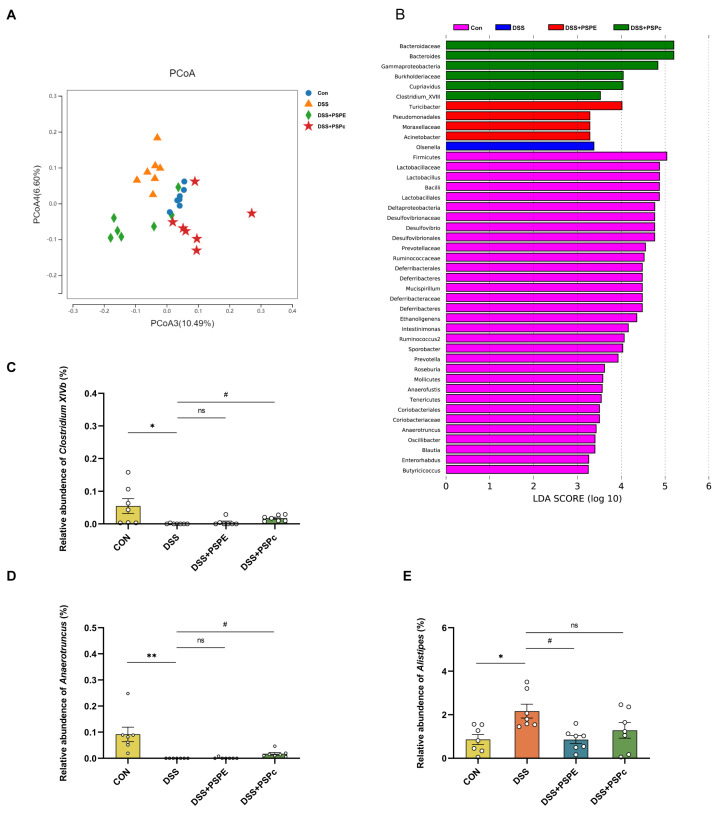
Effect of PSPE and PSPc on the gut microbiome compositions in DSS-induced colitis mice. (**A**) Principal components analysis (PCoA); (**B**) linear discriminant snalysis (LDA) score. (**C**) Relative abundance (%) of *Clostridium XlVb*; (**D**) relative abundance (%) of *Anaerotruncus*; (**E**) relative abundance (%) of *Alistipes*; data are presented as mean ± SE, and statistical significance was determined by Kruskal–Wallis test (H test) with, *n* = 7 mice. * *p* < 0.05, ** *p* < 0.01, compared with the Con group, # *p* < 0.05, compared with the DSS group, ns = not significant.

**Figure 6 antioxidants-11-02098-f006:**
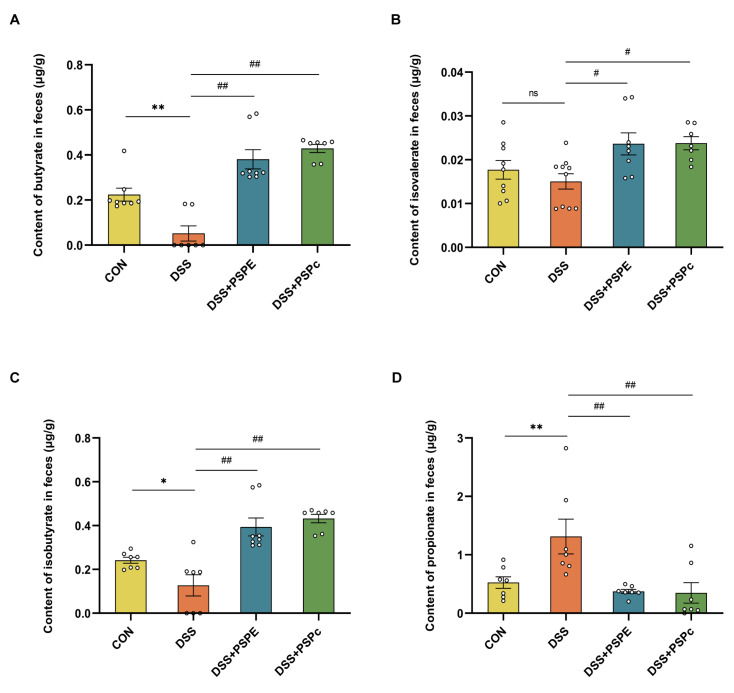
Effect of PSPE and PSPc on the SCFA generation in DSS-induced colitis mice. (**A**) Content of butyrate in feces; (**B**) content of isobutyrate in feces; (**C**) content of isovalerate in feces; (**D**) content of propionate in feces; data were expressed as mean ± SE (*n* = 7–10 mice). * *p* < 0.05, ** *p* < 0.01, compared with the Con group, # *p* < 0.05, ## *p* < 0.01, compared with the DSS group, ns = not significant.

**Figure 7 antioxidants-11-02098-f007:**
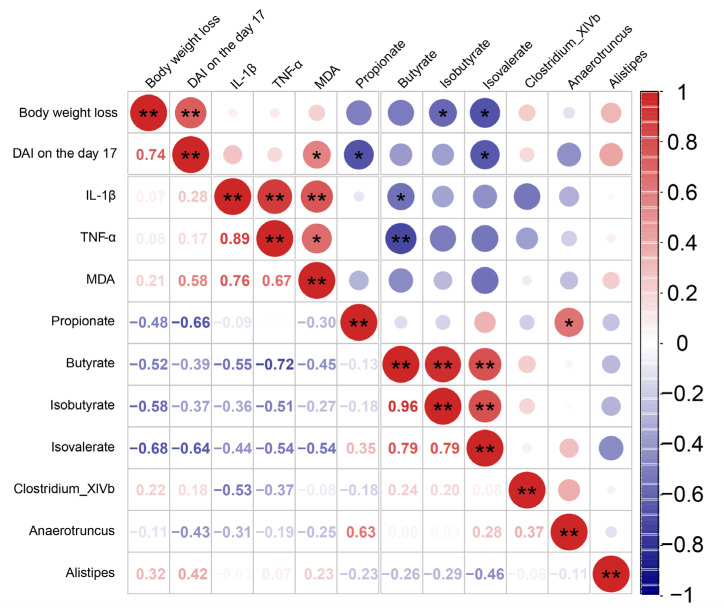
Correlation analysis among the DAI, fecal SCFAs levels, colon IL-1β, TNF-α, and MDA levels, and gut microbiota. Spearman’s correlation analysis was performed. In the upper right corner, you could find the size and color of the circle used to represent the relevant indicator level, with red indicating positive and blue indicating negative. The corresponding value of the corresponding index could be found in the lower left corner. Correlation analysis among indicators,* *p* < 0.05, ** *p* < 0.01.

**Table 1 antioxidants-11-02098-t001:** Primer sequences used for qRT-PCR analysis.

	Forward Primers (5′-3′)	Reverse Primer (5′-3′)
Occludin	ACGGACCCTGACCACTATGA	TCAGCAGCAGCCATGTACTC
IL-6	CTCTGGCGGAGCTATTGAGA	AAGTCTCCTGCGTGGAGAAA
iNOS	GGGCTGACCTGTTTCCTACT	GGAGGTTGAGACCCAATGGA
COX-2	CCCATTAGCAGCCAGTTGTC	CAGGATGCAGTGCTGAGTTC
Claudin-1	AGCTGCCTGTTCCATGTACT	CTCCCATTTGTCTGCTGCTC
MUC2	AGGGCTCGGAACTCCAGAAA	CCAGGGAATCGGTAGACATCG
Gapdh	TGGAGAAACCTGCCAAGTATGA	TGGAAGAATGGGAGTTGCTGT

**Table 2 antioxidants-11-02098-t002:** Mass spectrometry information of the main compounds in PSPc.

Compound	Matter	Rt (min)	MS[M-H] (m/z)	MS/MS (m/z)
1	Protocatechuic acid	7.388	153	109.03
2	B-type procyanidins	8.286	577	125.02, 287.05, 289.07, 407.07, 425.08, 451.10
3	Protocatechualdehyde	9.052	137	119.01, 108.02
4	A-type procyanidins dimer	10.665	575	285.04, 289.07, 407.07, 449.08
5	Catechins	11.741	289	125.02, 137.02, 165.02, 179.03, 205.05, 245.08
6	A-type procyanidins dimer	12.279	575	285.04, 289.07, 407.07, 449.08
7	A-type procyanidins trimer	13.648	863	287.05, 289.07, 411.07, 451.07, 575.12, 711.13
8	A-type procyanidins dimer	14.110	575	285.04, 289.07, 407.07, 423.07, 49.08
9	A-type procyanidins tetramer	16.126	1149	573.10, 575.12, 997.17

## Data Availability

Data is contained within the article and Appendix A.

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
