# Peer review of "The Peanut Skin Procyanidins Attenuate DSS-Induced Ulcerative Colitis in C57BL/6 Mice"

_antioxidants, 2022, doi:10.3390/antiox11112098_

Round 1

Reviewer 1 Report (New Reviewer)

The paper is very interesting and obtained results may have some important pratical meaning. However, some parts of the manuscript must be improved - especially Materials and methods section must be better described.

1. In the Abstract please explain the abbreviation DAI. 

2. line 63 - "is one of the most" - it should be "are one of the most", as there are many and not one substance from this group

3. line 124 - "PSPc was determined by HPLC" - what does it exactly mean? What HPLC has shown? Later there is infromation that the complex was analyzed by UPLC-QTOF-MS/MS. So, why this equipment was not desribed in Materials and methods chapter? What were chromatographic conditions for UPLC-MS analysis? Why it was not described?

4. Chapter 2.2 Animal experiment - how many animals were in each group? How PSPc was administarted? It was dissolved in water? In saline? How many times each day, PSPc was administrated? Why the experiment was conducted for 17 days? It was arbitrally chosen or based on some literature data? This whole part must be improved.

5. line 433 - please remove "the"

6. Paper presents interesting findings. However, what practical meaning it may have? People do not eat peanut skin. Do the Authors propose to use this extract or purified PSPc as nutraceuticals (eg. in the production of dietary supplements or fortified foods)? It must be clearly stated at the end of discussion.

Author Response

Dear Editor and Reviewers,

We sincerely appreciate for your constructive and helpful comments and suggestions to our manuscript entitled “The peanut skin procyanidins attenuate DSS-induced ulcera-tive colitis in C57BL/6 mice”, Manuscript ID: antioxidants-1905679. We have carefully modified our manuscript based on comments, and the major corrections/revisions are listed below, point-by-point. These changes will not affect the content and framework of the manuscript. We hoped that the revised manuscript would meet the publication standards of antioxidants. If there are any questions or problems about our manuscript, please don't hesitate to let us know.

Reviewer 1 comments:

The paper is very interesting and obtained results may have some important pratical meaning. However, some parts of the manuscript must be improved - especially Materials and methods section must be better described.

Comment 1: In the Abstract please explain the abbreviation DAI.

Response: Thanks very much for your suggestion. The abbreviation DAI have explained in the Abstract.

Revised as:

Line 22-23: The results showed that both PSPE and PSPc supplementation reversed the DSS-induced body weight loss and reduced disease activity index (DAI) values.

Comment 2: line 63 - "is one of the most" - it should be "are one of the most", as there are many and not one substance from this group.

Response: Thanks very much for your suggestion. We have revised in the manuscript.

Revised as:

Line 62-64: Peanut skin procyanidins (PSPc), which amounts to 128 g/kg [15], are one of the most pivotal and functional polyphenols in peanut skin.

Comment 3: line 124 - "PSPc was determined by HPLC" - what does it exactly mean? What HPLC has shown? Later there is information that the complex was analyzed by UPLC-QTOF-MS/MS. So, why this equipment was not desribed in Materials and methods chapter? What were chromatographic conditions for UPLC-MS analysis? Why it was not described?

Response: Thanks very much for your suggestion. In this study, the purity of PSPc was determined and analyzed by HPLC, and its components were identified by UPLC-QTOF-MS/MS. This was done earlier in the laboratory and, therefore, has only been described briefly. Now we have carefully added the operating equipment and methods in the manuscript.

Revised as:

Line 127-137: PSPc was determined by HPLC (Thermo Scientific UltiMate 3000 U, Phenomenex Kinetex C18) with a purity at ca. 95 %. The composition of PSPc was detected by ultrahigh-performance liquid chromatography-tandem mass spectrometry (UPLC-QTOF-MS/MS) (Agilent Technologies Technology (China) Co., Ltd.). The operating conditions of UPLC-QTOF-MS/MS were as follows: waters BEH C18 column (2.1×100 mm, 1.7 µm), mobile phase A: 0.1% formic acid in water; Mobile phase B: acetonitrile solution, gradient elution: 0min, 95%A, 5%B; 30min, 10%A, 90%B; 45min, 0%A, 100%B; 50min, 0%A, 100%B; 51min, 95%A, 5%B; 60min, 95%A, 5%B. The real-time flow rate is 0.3mL/min, the injection volume is 5µl, the real-time mass spectrometry scanning range is 50~1200 m/z, the sheath gas temp 350, the sheath gas flow 12L/min, the ESI-mode and the voltage are 3200V.

Comment 4: Chapter 2.2 Animal experiment - how many animals were in each group? How PSPc was administarted? It was dissolved in water? In saline? How many times each day, PSPc was administrated? Why the experiment was conducted for 17 days? It was arbitrally chosen or based on some literature data? This whole part must be improved.

Response: Thanks very much for your suggestion. We have revised in the manuscript as your suggestion.

Revised as:

Line 144-149:The mice were divided into four groups randomly: the control group (n=10), the DSS group (n=10), the DSS+PSPE group (n=10), and the DSS+PSPc group (n=10). The control group and the DSS group were treated with physiological saline for 17 days by gavage before euthanasia. The DSS+PSPE group (200 mg/kg of PSPE dissolved in saline) and the DSS+PSPc group (200 mg/kg of PSPc dissolved in saline) were administered with corresponding doses by gavage once daily for 17 days [33,34].

Comment 5: line 433 - please remove "the".

Response: Thanks very much for your suggestion. We have revised in the manuscript.

Comment 6: Paper presents interesting findings. However, what practical meaning it may have? People do not eat peanut skin. Do the Authors propose to use this extract or purified PSPc as nutraceuticals (eg. in the production of dietary supplements or fortified foods)? It must be clearly stated at the end of discussion.

Response: Thanks very much for your suggestion. We have revised in the manuscript.

Revised as:

Line 530-532:It suggests that purified PSPc have potential as dietary supplements or for the production of fortified foods as a promising nutritional strategy for reducing UC.

Reviewer 2 Report (New Reviewer)

The researchers conducted a thorough study on the effects of peanut skin extract and procyanidins on DSS-induced colitis. Several different methodological approaches were used to provide a comprehensive look at the impact of PSPE and PSPC on gut microbiota, colonic histology, and colonic inflammation. Several minor issues can be addressed that would strengthen the paper:

·      Line 67-69. The wording here should be softened, and more appropriate references should be cited. This sentence claims that PSPc have many beneficial functions for human health, including anti-allergy, anti-cancer, etc. Two sources are cited here. One appears to be a characterization of procyanidins from peanut skins and the other is a mouse study of atherosclerosis. Please clarify if most of the work has been done in humans or preclinical models, and cite sources to support the myriad of conditions mentioned to be benefitted by PSPc in this sentence.

·      2.2 Animal Experiment: Please clarify the sex of the mice.

·      Line 171: Please clarify the expertise of the “experienced staff” assessing the H&E staining and whether these staffers were blinded to the samples’ treatments.

·      Immunohistochemistry and qRT-PCRmethods: rather than providing volumes of antibodies (i.e. 50 uL primary; 1:300 dilution), please provide the concentrations that were used.

·      Line 205: what is meant by “pore sample”?

·      Line 206: What “culture supernatant” is being referred to here?

·      Figure 2: please include a figure legend for the line graphs indicating which color line corresponds to which treatment group. The authors may also want to consider change the two shades of green to be more distinct. Looking at the line graphs, it is difficult to distinguish the two shades of green.

·      Figure 2C: please confirm that the units on the y-axis are correct. It looks like % body weight loss would be closer to 15% for the DSS group, not .15 or whatever is indicated on the figure.

·      Some grammatical suggestions

o   Line 23: spell out “DAI” in the abstract

o   Line 41: do not capitalize “It’s”

o   Line 65: no need to abbreviate “ATD” and “BTD” since these abbreviations are not used later.

o   Line 81: I think “integrality” is supposed to be “integrity”.

o   Line 89: do not capitalize “Various”.

o   Line 96 and line 104: Abbreviate SCFAs earlier as the abbreviation is used earlier. Once it is spelled out and abbreviated, there is no need to repeat this abbreviation.

o   Line 168: no need to spell out “H&E” again since it is already spelled out and abbreviated in line 166.

Line 181: Should “Carnot” be “carnoy”?

Author Response

Dear Editor and Reviewers,

We sincerely appreciate for your constructive and helpful comments and suggestions to our manuscript entitled “The peanut skin procyanidins attenuate DSS-induced ulcera-tive colitis in C57BL/6 mice”, Manuscript ID: antioxidants-1905679. We have carefully modified our manuscript based on comments, and the major corrections/revisions are listed below, point-by-point. These changes will not affect the content and framework of the manuscript. We hoped that the revised manuscript would meet the publication standards of antioxidants. If there are any questions or problems about our manuscript, please don't hesitate to let us know.

Reviewer comments:

The researchers conducted a thorough study on the effects of peanut skin extract and procyanidins on DSS-induced colitis. Several different methodological approaches were used to provide a comprehensive look at the impact of PSPE and PSPC on gut microbiota, colonic histology, and colonic inflammation. Several minor issues can be addressed that would strengthen the paper:

Comment 1: Line 67-69. The wording here should be softened, and more appropriate references should be cited. This sentence claims that PSPc have many beneficial functions for human health, including anti-allergy, anti-cancer, etc. Two sources are cited here. One appears to be a characterization of procyanidins from peanut skins and the other is a mouse study of atherosclerosis. Please clarify if most of the work has been done in humans or preclinical models, and cite sources to support the myriad of conditions mentioned to be benefitted by PSPc in this sentence.

Response: Thanks very much for your suggestion. We have added more cite sources to support the myriad of conditions mentioned to be benefitted by PSPc. 

Revised as:

Line 67-70: Moreover, accoding to many in vitro and in vivo studies, PSPc is shown to have many potentially beneficial functions for human health, including anti-oxidative activities, antibacteria, anticancer, hyperlipidemia prevention, anti-allergy, hypoglycemic capability, and inhibiting acrylamide production [18–24]. 

Comment 2: 2.2 Animal Experiment: Please clarify the sex of the mice.

Response: Thank you for your question. The male mice (C57BL/6J, 8-weeks-old, n=40) were used in this study.

Revised as:

Line 132: The male mice (C57BL/6J, 8-weeks-old, n=40) were purchased from Xi’an Jiaotong University (Xi’an, Shaanxi, China).

Comment 3: Line 171: Please clarify the expertise of the “experienced staff” assessing the H&E staining and whether these staffers were blinded to the samples’ treatments.

Response: Thank you for your question. In our study, the staff engaged in tissue section staining technology have received professional training and had a lot of practical experience. Histological scores were assessed on the basis of the extent of inflammatory cell infiltration and mucosal damage by these professionals, who were blinded to the samples’ treatments.

Revised as:

Line 174-177: The histological scores were assessed by experienced staff in the laboratory according to the extent of infiltration of inflammatory cells and mucosa damage, and these staffers had a lot of hands-on experience, but they were blinded to the samples’ treatments.

Comment 4: Immunohistochemistry and qRT-PCR methods: rather than providing volumes of antibodies (i.e. 50 uL primary; 1:300 dilution), please provide the concentrations that were used.

Response: Thank you for your question. While immunohistochemistry and qRT-PCR analyses were assessed by kit method, it was regrettable that no description of antibodies concentration was provided in the kit.  

Comment 5: Line 205: what is meant by “pore sample”?

Response: Thank you for your question. The “pore sample” meant a sample in each well of a porous plate.

Comment 6: Line 206: What “culture supernatant” is being referred to here?

Response: Thank you for your question. It was an error. The supernatant was the product of colon tissue homogenization and centrifugation. We revised the paragraph in the manuscript.

Revised as:

Colon tissues were homogenized, centrifuged, and attained the supernatant to analyze cytokines. Proinflammatory cytokine concentrations (MDA, TNF-α, and IL-1β) were quantified by using ELISA kits (Xinle Biotechnology, Shanghai, China), and measured at an optical density of 450 nm (Bio-Rad, CA, USA).

Comment 7: Figure 2: please include a figure legend for the line graphs indicating which color line corresponds to which treatment group. The authors may also want to consider change the two shades of green to be more distinct. Looking at the line graphs, it is difficult to distinguish the two shades of green.

Response: Thank you for your question. We've tweaked Figure 2 to make it look clearer and more accurate.  

Comment 8: Figure 2C: please confirm that the units on the y-axis are correct. It looks like % body weight loss would be closer to 15% for the DSS group, not .15 or whatever is indicated on the figure.

Response: Thank you for your question. We've tweaked Figure 2C to make it look clearer and more accurate. 

Comment 9:  Some grammatical suggestions:

Line 23: spell out “DAI” in the abstract

Line 41: do not capitalize “It’s”

Line 65: no need to abbreviate “ATD” and “BTD” since these abbreviations are not used later.

Line 81: I think “integrality” is supposed to be “integrity”.

Line 89: do not capitalize “Various”.

Line 96 and line 104: Abbreviate SCFAs earlier as the abbreviation is used earlier. Once it is spelled out and abbreviated, there is no need to repeat this abbreviation.

Line 168: no need to spell out “H&E” again since it is already spelled out and abbreviated in line 166.

Line 181: Should “Carnot” be “carnoy”?

Response: Thank you very much for putting forward such detailed modification suggestions to our article. We have carefully checked the language and grammar of the full text, and revised all the problems pointed out above. 

This manuscript is a resubmission of an earlier submission. The following is a list of the peer review reports and author responses from that submission.

Round 1

Reviewer 1 Report

The manuscript “The peanut skin procyanidins attenuate DSS-induced ulcerative colitis in C57BL/6 mice” by Na Wang et al. They have reported the beneficial effects of peanut skin procyanidins against DSS-induced ulcerative colitis using a mice model. Authors have done biochemical assays, ELISA, Immunohistochemical, gas chromatograph, and histological analyses, to prove their hypothesis. The manuscript is not written well it has several grammatical and typographical errors. After thoroughly reviewing I feel the manuscript needs to be extensively revision

Comments:

1.     Section 2.2 Animal experiment, is confusing, need to rename the groups properly.

2.     In section 2.6 Immunohistochemistry analysis, authors should specify the primary and secondary antibody used and their dilution.

3.     Section 2.8 qRT-PCR analysis, is written in a very confusing way. Authors must specify the concentrations of all the components used. Authors have mentioned only the volume in layman style. The authors should also have mentioned the amount of RNA used to synthesize cDNA. The author should also have mentioned the source of primers whether it was in-house designed?

4.     In figure 3A-C magnification is missing, authors must add it.

5.     The letters used to denote the significant difference are confusing, authors must change this.

6.     The authors should check grammatical errors very carefully and the writing style must be improved.

7.     I will suggest explaining properly the discussion section.

8.     I will suggest adding a graphical abstract, which is the point of attraction for a reader.

Reviewer 2 Report

The study “The peanut skin procyanidins attenuate DSS-induced ulcera-tive colitis in C57BL/6 mice” by Wang et al is presented for peer review.

- Graphs are not clear, change color and show individual data points

- indicate the level of significance in the figure legend

- Who did the histological scoring? A pathologist?

- Include zoom-in pictures for Figure 3 A/B/C

- add stars, asterisk etc to highlight changes

- provide additional analysis of microbiome in germ free mice

- I don’t see any prove of mechanism. Please provide intracellular pathways